# Spin-induced multiferroicity in the binary perovskite manganite $Mn_2O_3$

Junzhuang Cong[1], Kun Zhai[1], Yisheng Chai[1], Dashan Shang [1], Dmitry D. Khalyavin[2], Roger D. Johnson[3], Denis P. Kozlenko[4], Sergey E. Kichanov[4], Artem M. Abakumov[5], Alexander A. Tsirlin[6], Leonid Dubrovinsky[7], Xueli Xu[8], Zhigao Sheng [8], Sergey V. Ovsyannikov[7,9] & Young Sun [1,10]

The $ABO_3$ perovskite oxides exhibit a wide range of interesting physical phenomena remaining in the focus of extensive scientific investigations and various industrial applications. In order to form a perovskite structure, the cations occupying the $A$ and $B$ positions in the lattice, as a rule, should be different. Nevertheless, the unique binary perovskite manganite $Mn_2O_3$ containing the same element in both $A$ and $B$ positions can be synthesized under high-pressure high-temperature conditions. Here, we show that this material exhibits magnetically driven ferroelectricity and a pronounced magnetoelectric effect at low temperatures. Neutron powder diffraction revealed two intricate antiferromagnetic structures below 100 K, driven by a strong interplay between spin, charge, and orbital degrees of freedom. The peculiar multiferroicity in the $Mn_2O_3$ perovskite is ascribed to a combined effect involving several mechanisms. Our work demonstrates the potential of binary perovskite oxides for creating materials with highly promising electric and magnetic properties.

[1] Beijing National Laboratory for Condensed Matter Physics, Institute of Physics, Chinese Academy of Sciences, Beijing 100190, China. [2] ISIS Facility, Rutherford Appleton Laboratory-STFC, Chilton, Didcot OX11 0QX, UK. [3] Department of Physics, Clarendon Laboratory, University of Oxford, Oxford OX1 3PU, UK. [4] Frank Laboratory of Neutron Physics, Joint Institute for Nuclear Research, 141980 Dubna, Russia. [5] Center for Electrochemical Energy Storage, Skolkovo Institute of Science and Technology, Nobel Street 3, 143026 Moscow, Russia. [6] Experimental Physics VI, Center for Electronic Correlations and Magnetism, Institute of Physics, University of Augsburg, 86135 Augsburg, Germany. [7] Bayerisches Geoinstitut, Universität Bayreuth, Universitätsstrasse 30, 95447 Bayreuth, Germany. [8] High Magnetic Field Laboratory, Chinese Academy of Sciences, Hefei, Anhui 230031, China. [9] Institute for Solid State Chemistry, Russian Academy of Sciences, Urals Division, 91 Pervomayskaya Str., Yekaterinburg 620990, Russia. [10] School of Physical Sciences, University of Chinese Academy of Sciences, Beijing 100190, China. Correspondence and requests for materials should be addressed to S.V.O. (email: sergey.ovsyannikov@uni-bayreuth.de) or to Y.S. (email: youngsun@iphy.ac.cn)

Perovskite structures of transition-metal oxides provide an unprecedented playground for condensed-matter physics and materials science. They exhibit numerous exciting phenomena and functional properties, such as exotic magnetism, ferroelectricity, high-$T_c$ superconductivity, colossal magnetoresistance, charge/orbital ordering, significant thermoelectricity, giant negative thermal expansion, photoelectricity, multiferroicity, etc[1–5]. There are two different cation sites in the $ABO_3$ perovskite structure, namely, $B$-sites that are octahedrally coordinated by oxygen and $A$-sites that are cuboctahedrally coordinated and often heavily distorted[6]. Small and highly charged transition-metal cations normally occupy the $B$-sites, but they are hardly suitable to occupy the large 12-coordinated $A$-sites. A partial filling of the $A$-positions with transition metals is possible in a unique class of $A$-site-ordered perovskites, e.g., in such manganites as $(AMn_3)Mn_4O_{12}$ ($A$ = Na, Ca, La, Bi)[7–10]. The full occupation of the $A$-sites with transition metals would yield a $B_2O_3$ binary perovskite, but until recently this remained a formidable challenge.

Recently, the first binary perovskite oxide, $Mn_2O_3$, was synthesized under high-pressure and high-temperature (HP-HT) conditions[11,12]. It is worth mentioning that it was found by a single-crystal X-ray diffraction technique that $Fe_2O_3$ also adopts a perovskite structure at HP-HT conditions being, however, not quenchable to ambient pressure[13]. The Mn cations show variable charge states (e.g., $Mn^{2+}$, $Mn^{3+}$, and $Mn^{4+}$) with ionic radii and, in particular, the $Mn^{2+}$ cation is sufficiently large to be accommodated in the $A$-position of the perovskite structure. Using synchrotron X-ray diffraction (SXRD), electron diffraction, annular bright-field scanning transmission electron microscopy (ABF-STEM), and electron energy loss spectroscopy (EELS)[11,12], we demonstrated that the perovskite-type $Mn_2O_3$ can be represented as $(Mn^{2+}Mn^{3+}_3)Mn^{3.25+}_4O_{12}$, given different oxidation states of Mn depending on the crystallographic position.

As shown in Fig. 1, the crystal structure of perovskite-type $Mn_2O_3$ is determined within the $a = 4a_p$ supercell[11]. It is a distorted variant of the $AA'_3B_4O_{12}$ quadruple perovskites with an in-phase $a^+b^+c^+$ octahedral tilt according to the Glazer notation[14]. As all the $A$-sites are occupied by the Mn ions, the octahedral tilt is significant, reaching about 15–18° (Fig. 1a). The $Mn_B$–O interatomic distances are in the range of 1.82–2.20 Å so that the $Mn_BO_6$ octahedra are significantly deformed. A distorted square-planar first coordination sphere for the $Mn_{A'}$ cations with the $Mn_{A'}$-O distance in the range of 1.81–2.09 Å is created due to the cooperative tilt of the $Mn_BO_6$ octahedra. The second coordination sphere of the $Mn_{A'}$ cations includes one to two oxygen atoms positioned at much larger distances of 2.33–2.37 Å. Consequently, the complete coordination environment is a distorted tetragonal pyramid or octahedron. Due to the rotation and distortion of the $Mn_BO_6$ octahedra, a distorted octahedral environment for the $Mn_A$ cations with the $Mn_A$–O distances in the range of 2.24–2.69 Å is created.

One unique feature of this binary perovskite $Mn_2O_3$ is that all the $A$- and $B$-site cations can be magnetically and electrically active, thus leading to an interplay of multiple structural and electronic instabilities[15,16]. Besides the conventional $Mn_B$–O–$Mn_B$ interaction path, the binary perovskite has additional paths through the A-cations along the chains of $Mn_{A'}$–O–$Mn_B$, $Mn_A$–O–$Mn_B$, $Mn_A$–O–$Mn_{A'}$, and $Mn_{A'}$–O–$Mn_A'$. The highly distorted and tilted structure along with the competing multiple interactions could induce unexpected interplay of different properties in the binary perovskite $Mn_2O_3$, compared to those known in conventional perovskite oxides. More details of the crystal structure of the perovskite-type $Mn_2O_3$ may be found in earlier publication[11].

In this work, we report the observation of a spin-induced multiferroicity in this unique $Mn_2O_3$ perovskite. The coexistence

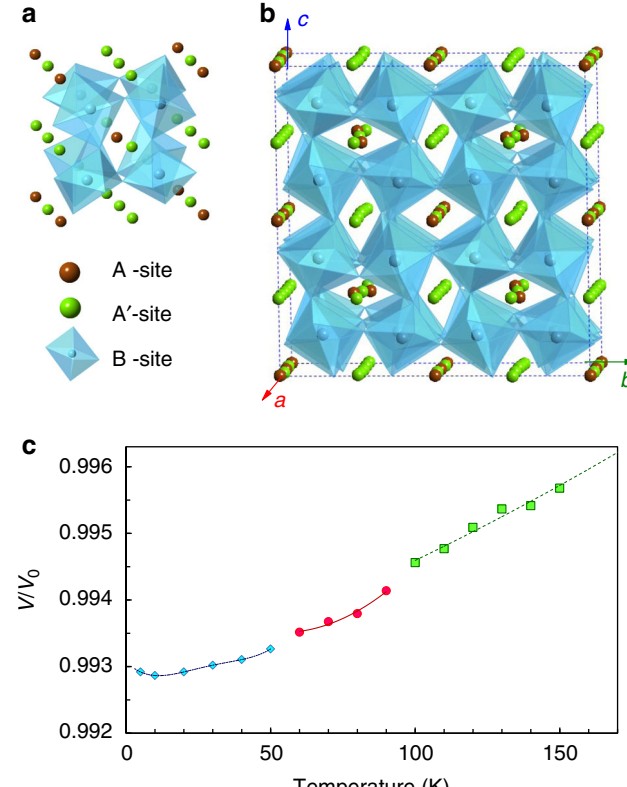

**Fig. 1** Crystal structure of the perovskite-type $Mn_2O_3$. **a** A highly distorted and tilted perovskite unit cell. **b** The crystal structure of the binary perovskite $Mn_2O_3$ determined within the $a = 4a_p$ supercell. The oxygen atoms at the corners of the $Mn_BO_6$ octahedra are omitted for clarity. It is a distorted variant of the $AA'_3B_4O_{12}$ perovskite with an in-phase cooperative $a^+b^+c^+$ octahedral tilt. **c** Relative thermal expansion of the perovskite $Mn_2O_3$ ($V_0$ is the unit cell volume at ambient conditions). One can see the weak volumetric changes upon the magnetic transitions

of magnetism and ferroelectricity in perovskites, such as $BiFeO_3$ (ref. [17]) and $TbMnO_3$ (ref. [18]), renders such compounds an interesting field of research and creates potential for industrial applications[19–21]. In the last decade, a large number of multiferroic materials with different mechanisms of magnetoelectric (ME) coupling were discovered. However, binary oxides showing multiferroicity are rare. CuO (refs. [22–24]), $Fe_3O_4$ (ref. [25]), and $Fe_2O_3$ (ref. [26]) are the few known examples reported so far. $Mn_2O_3$ represents the first example of multiferroic binary oxide with the unusual perovskite structure.

## Results

**Multiple magnetic transitions in the binary perovskite $Mn_2O_3$.** The magnetic susceptibility ($\chi$) of the $Mn_2O_3$ perovskite measured in a low magnetic field ($H = 500$ Oe) is presented in Fig. 2a. At temperatures above ~ 180 K, it follows the Curie–Weiss law $\chi = C/(T\text{-}\theta)$ with the Curie–Weiss temperature $\theta = -327$ K. The negative $\theta$ value suggests predominantly antiferromagnetic (AFM) interactions between the Mn ions. At low temperatures, the $Mn_2O_3$ perovskite undergoes two magnetic transitions at $T_1 = 101$ K and $T_2 = 49$ K, as further confirmed by heat capacity measurements. Since the magnetic ordering occurs at temperatures much lower than $|\theta|$, strong magnetic frustration is present in the system. The drops of $\chi$ with decreasing temperature reflect the AFM nature of both transitions (Fig. 2). However, the magnitudes of these dips are small, and the magnetic susceptibility $\chi$

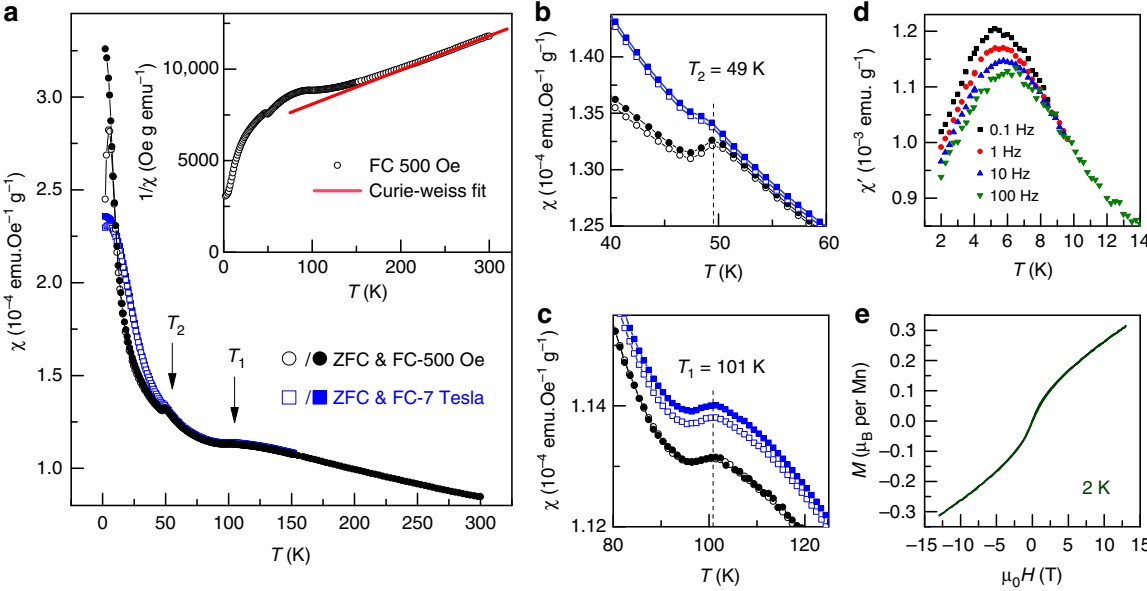

**Fig. 2** Magnetic properties of the perovskite-type $Mn_2O_3$. **a** Temperature dependence of DC magnetic susceptibility measured in the field of 500 Oe with both the zero-field-cooling (ZFC) and field-cooling (FC) modes. This curve distinctly demonstrates two magnetic transitions at $T_1 = 101$ K and $T_2 = 49$ K. The inset shows a Curie–Weiss fit of the paramagnetic susceptibility. **b**, and **c** shows an effect of applied magnetic field on the above-mentioned magnetic transitions. While the transition at $T_2$ is apparently suppressed by application of a 7 T field, the transition at $T_1$ is very weakly affected by 7 T field. **d** AC magnetic susceptibility as a function of temperature. The frequency dependent peak around 5 K suggests a short-range ordered magnetic state. **e** The M–H curve at 2 K showing no saturation of magnetization up to 13 T

starts increasing again only a few Kelvin below $T_1$ ($T_2$). These features suggest that only a partial magnetic ordering occurs at both transitions. A group of the magnetic Mn cations may remain disordered because of the frustration and become randomly frozen at low temperatures, as indicated by a divergence between the ZFC and FC susceptibility curves below ~ 5 K. Furthermore, the low-temperature transition at $T_2$ is clearly suppressed in a high magnetic field (Fig. 2b). In contrast, the high-temperature transition at $T_1$ is not affected even by a strong magnetic field of 7 T (Fig. 2c). The AC magnetic susceptibility exhibits a frequency-dependent peak around 5 K (Fig. 2d) and the magnetization at 2 K is far from saturation up to 12 T (Fig. 2e). These features further suggest a short-ranged ordered magnetic state at low temperatures.

In order to examine structural changes upon the low-temperature magnetic transitions, we carried out high-resolution powder X-ray diffraction studies at the ID22 beamline of the European Synchrotron Radiation Facility (Grenoble, France) at temperatures below 150 K. Rietveld refinements of these XRD patterns showed the persistence of the original room-temperature structure across the transitions. At the lowest temperature of 5 K we reached in these experiments, we found the same face-centered triclinic $F\bar{1}$ structure with the unit cell parameters $a = 14.6505(5)$ Å, $b = 14.6300(5)$ Å, $c = 14.6394(7)$ Å, $\alpha = 89.306(4)°$, $\beta = 89.288(4)°$, $\gamma = 89.293(7)°$, $V = 3137.1(1)$ Å³, and $Z = 64$. Temperature evolution of the unit cell volume of perovskite $Mn_2O_3$ demonstrates the anomalies at the magnetic transitions (Fig. 1c). One can see that the magnetic ordering leads to minor volumetric effects, indicating the presence of magnetoelastic coupling.

**Neutron powder diffraction data**. Further insights into the nature of the magnetically ordered states of perovskite-type $Mn_2O_3$ are gained from neutron powder diffraction (NPD) data shown in Fig. 3a, b. Due to a complex character of the triclinic $F\bar{1}$ crystal structure with 20 symmetry independent magnetic

sites per unit cell, solution of the magnetic structures is an extremely challenging problem. To impose some symmetry restrictions, we adapted an approach, which uses the higher-symmetry $R\bar{3}$ approximation for the crystal structure and is based on the knowledge of the magnetic structures in closely related $A^{2+}Mn_7O_{12}$ ($A^{2+}$ = Ca, Sr, Pb and Cd) quadruple manganites[27–29]. This approach is well justified, owing to the fact that the triclinic distortion is very small[11] and could not be resolved in our high-resolution neutron powder diffraction data. The trigonal $R\bar{3}$ space group with a large unit cell $a \approx 2\sqrt{2}a_p$ and $c \approx 4\sqrt{3}a_p$ is an isotropy subgroup, associated with the $\mathbf{k}_S = (0,0,3/4)$ propagation vector[30,31] of the $R\bar{3}$ group, having a smaller unit cell size $a \approx 2\sqrt{2}a_p$ and $c \approx 4\sqrt{3}a_p$. The latter structural model was used as the parent structure for the magnetic structure solutions and Landau free-energy decomposition in the $A^{2+}Mn_7O_{12}$ manganites[27–29]. We also used the $R\bar{3}$ ($a \approx 2\sqrt{2}a_p$, $c \approx 4\sqrt{3}a_p$) symmetry in our analysis, which allowed us to drastically reduce the number of symmetry unrelated sites and resulted in successful determination of the spin structures in both magnetic phases of $Mn_2O_3$ (Fig. 4).

In the temperature range of 49 K < T < 101 K, the magnetic structure is a longitudinal spin-density wave (Fig. 4a) with the propagation vector $\mathbf{k}_0 = (0,0,9/8)$ locked to the structural modulation through the relation $\mathbf{k}_S + 2\mathbf{k}_0 = 3$. It is remarkable that, apart from the fundamental $\mathbf{k}_0$ component, a presence of a sizable third harmonic $3\mathbf{k}_0 \equiv (0,0,3/8)$ is clearly visible (Fig. 3a), revealing essentially anharmonic character of the modulated magnetic structure. The magnetic space group of this anharmonic spin-density wave is $R_I\bar{3}$, which implies a non-polar nature of the lock-in magnetic phase. The corresponding Fourier coefficients for the four symmetry independent Mn sites in the parent structure, refined from the data collected at $T = 60$ K, are given in Supplementary Table 1. In addition, the magnetic structure details of this high-temperature phase are presented in Supplementary Figure 1.

Below $T_2$, the magnetic propagation vector $\mathbf{k}_0$ de-locks from the structural modulation giving rise to an incommensurate

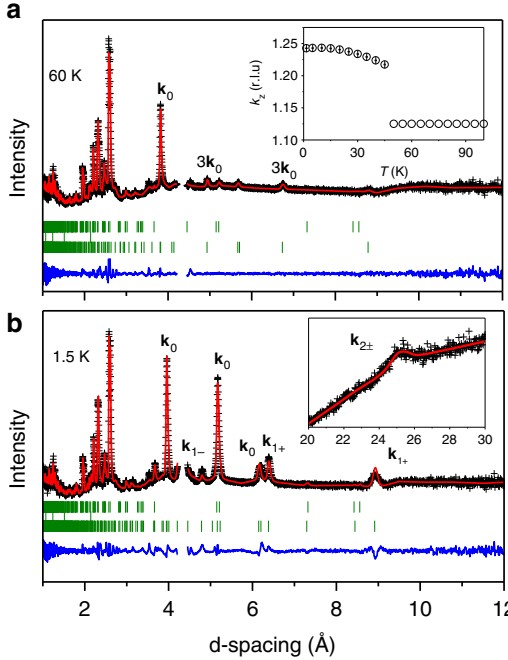

**Fig. 3** Neutron diffraction data of the perovskite-type $Mn_2O_3$. **a, b** Rietveld refinement of the WISH neutron diffraction patterns collected at 60 K and 1.5 K, respectively. The black cross symbols and red solid lines represent the experimental and calculated intensities, respectively. The blue lines at the bottom of both plots are the difference between the experimental and calculated intensities. There are three sets of tick marks: the upper one indicates the positions of Bragg peaks for the nuclear, the middle one shows a few reflections of the vanadium can for the sample, and the lower one indicates the positions of Bragg peaks for magnetic scattering. The inset in **a** shows temperature dependence of the $k_z$-component of the magnetic propagation vector $\mathbf{k}_0$. The inset in **b** shows magnified portion of the diffraction pattern where a satellite of the $\mathbf{k}_{2\pm}$ propagation vector is observed

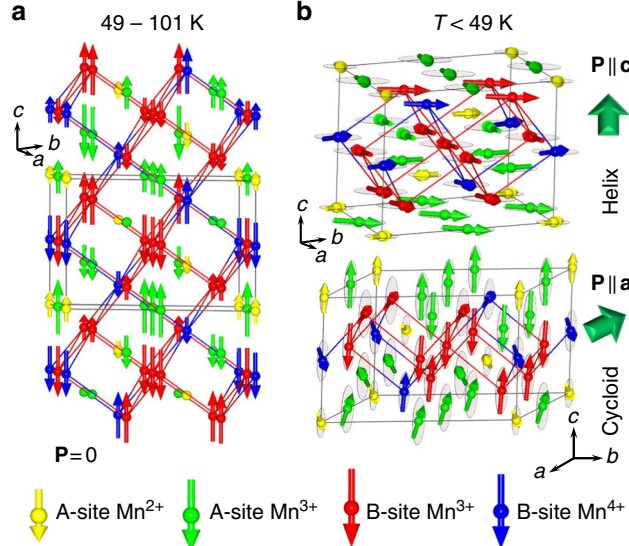

**Fig. 4** Magnetic structures of the perovskite-type $Mn_2O_3$. **a** Schematic representations of the magnetic structure in the commensurate high-temperature phase (49 K < T < 101 K). This structure is a longitudinal spin density wave with the unit cell eight times bigger than the cell of the parent $R\bar{3}$ ($2\sqrt{2}a_p \times 2\sqrt{2}a_p \times \sqrt{3}a_p$) structure. It combines two types of the B-site and three types of the A-site Mn-layers stacked along the c-axis. **b** Schematic representations of the magnetic structure in the incommensurate low-temperature phase (T < 49 K). This structure combines both cycloidal and helical components. **P** refers to the induced electric polarization

Factors that stabilize the complex magnetic structures in the binary perovskite $Mn_2O_3$ are probably common for the whole series of $A^{2+}Mn_7O_{12}$ manganites[28,29,32]. A competing exchange interactions of the Mn ions occupying both the A-site and B-site positions in the perovskite structure and modulated by the orbital density wave, play a central role in the stabilization of these magnetic structures. The main distinctive feature of the perovskite $Mn_2O_3$ are the direction of the magnetic moments in the high-temperature spin density wave, and the presence of the cycloidal component in the ground state. They both imply a different type of magnetic anisotropy, of which origin requires further experimental and theoretical studies.

**Spin-induced ferroelectricity in the binary perovskite $Mn_2O_3$.** The dielectric constant ($\varepsilon_r$) of the binary perovskite $Mn_2O_3$ as a function of temperature is shown in Fig. 5a. There is a clear dielectric peak with a thermal hysteresis at $T_2 \sim 49$ K. The position of the dielectric peak does not shift with frequency. These features indicate a first-order ferroelectric phase transition induced by the onset of the magnetic ordering at $T_2$. However, no apparent dielectric anomaly is detected around the magnetic transition at $T_1 = 101$ K. To confirm the ferroelectricity, the pyroelectric current measurements are performed from 2 K to 120 K with both positive and negative E-field poling (see Supplementary Figure 3). Two pyroelectric peaks are observed in the temperature range investigated. The pyroelectric peak around 49 K is consistent with the magnetic transition at $T_2$. When the poling electric field is reversed, the sign of pyroelectric peak is also reversed (Fig. 5b). Moreover, the position of this pyroelectric peak at $T_2$ does not shift with the heating rate (see Supplementary Figure 4). These features strongly suggest an intrinsic ferroelectricity below $T_2$. There is another pronounced peak around 80 K which does not coincide with any magnetic or structural phase transition and is independent of external magnetic fields (see Supplementary

ground state (Fig. 4b). Apart from $\mathbf{k}_0$, a set of additional magnetic reflections is observed, which can be indexed using propagation vectors from the series $\mathbf{k}_{n\pm} = \mathbf{k}_0 \pm n\mathbf{k}_S$ (n = 0, 1, 2…), similar to the multi-k ground state of $A^{2+}Mn_7O_{12}$ manganites[28,29]. This observation strongly supports the presence of 3:1 ($Mn^{3+}:Mn^{4+}$) charge order and orbital density wave in the system and reveals a crucial role of the magneto-orbital coupling[28,29,32]. The diffraction patterns provide evidence for the four distinct magnetic components, with $\mathbf{k}_0 = (0,0,1.2439(3))$, $\mathbf{k}_{1+} = (0,0,1.0060(3))$, $\mathbf{k}_{1-} = (0,0,0.4940(3))$ and $\mathbf{k}_{2+} = \mathbf{k}_{2-} (0,0,0.2560(3))$ at $T = 1.5$ K, which were combined to get a constant-moment phase-modulated cycloid (Fig. 4b). This remarkable magnetic structure results in a good refinement quality (Fig. 3) and naturally explains the multiferroic properties below $T_2$ (as described in the next section). The magnetic point group is triclinic 1' and sets no restrictions on the polarization direction. In addition, the symmetry allows an admixture of a helical component similar to the ground state of $A^{2+}Mn_7O_{12}$ manganites[28,29]. A tilting of the plane of the spin rotation by 20(2) degrees from the c-axis further improves the refinement quality, implying that the magnetic structure is very general and combines both the major cycloidal and the minor helical components (see Supplementary Table 2 for details). The above propagation vectors characterizing magnetic orders in the perovskite $Mn_2O_3$ are completely different from those found earlier in the cubic bixbyite $\alpha$-$Mn_2O_3$[33]. The magnetic structure details of this low-temperature phase are presented in Supplementary Figure 2.

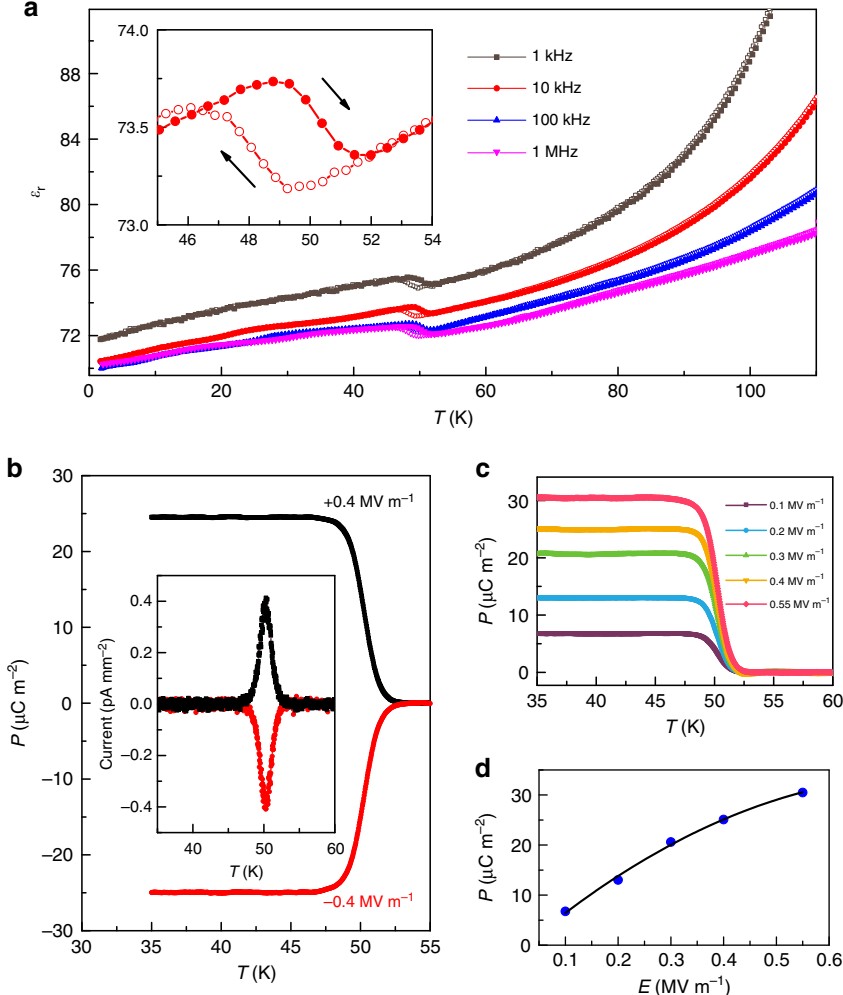

**Fig. 5** Spin-induced ferroelectricity in the binary perovskite $Mn_2O_3$. **a** The dielectric constant as a function of temperature. A dielectric peak with a hysteresis is seen at $T_2 = 49$ K, indicating a first-order ferroelectric transition. No apparent dielectric anomaly at $T_1 = 101$ K is detected. The inset shows an enlarged view around $T_2$. **b** The electric polarization as a function of temperature measured with a poling electric field of 0.4 MV m$^{-1}$. The polarization is reversed by a negative poling field. The inset presents the pyroelectric current as a function of temperature after subtracting the background. The peak around $T_2$ confirms spin-induced ferroelectricity. **c** The electric polarization measured with different poling electric field. **d** The electric polarization as a function of poling electric field

Figure 5). It could be due to extrinsic factors such as space trapped charges at grain boundaries. We note that a broad and high pyroelectric peak was often reported in the quadruple perovskite manganites such as $CaMn_7O_{12}$, and its origin is quite controversial[27,34–44].

To further clarify whether the pyroelectric peak at elevated temperatures corresponds to intrinsic ferroelectricity or not, we employ polarized second harmonic generation (SHG) technique with the 90° reflection geometry (see Supplementary Figure 7). The SHG signals obtained at high temperatures (120 and 70 K) are almost zero. In contrast, a non-zero SHG signal is visible at 40 K, which corroborates the emergence of the electric polarization at low temperatures. The SHG results further confirm that the broad peak of the pyroelectric current at high temperature (80 K) is due to extrinsic factors, whereas the spin-induced ferroelectricity below $T_2 = 49$ K is the intrinsic feature of our material. More importantly, these results for the perovskite $Mn_2O_3$ cast doubts concerning large polarization values which are often reported for quadruple perovskites of this family. For example, the giant ferroelectricity in $CaMn_7O_{12}$ reported previously could be delusive, whereas, intrinsic ferroelectricity below $T_1$ may be, in fact, rather weak.

The electric polarization (**P**) as a function of temperature is obtained by integrating the pyroelectric current with time after subtracting the background current (see Supplementary Figure 6 for more details). Figure 5c, d shows the poling electric field dependence of $P$. The value of the polarization increases steadily with increasing poling electric field, and reaches ~30 μC m$^{-2}$ at **E** = 0.55 MV m$^{-1}$.

In the low-temperature ferroelectric phase stabilizing below $T_2 = 49$ K, a pronounced ME effect is clearly observed. As shown in Fig. 6a, the pyroelectric peak around $T_2$ is gradually suppressed by increasing external magnetic field. This behavior is consistent with the magnetic susceptibility around $T_2$ (Fig. 2b), at which the magnetic transition is smeared out in high magnetic fields. Consequently, the electric polarization $P$ below $T_2$ decreases with increasing magnetic field (Fig. 6b), yielding a strong ME effect.

## Discussion
Below, we discuss a possible mechanism of spin-induced multiferroicity in the perovskite-type $Mn_2O_3$. The appearance of the ferroelectric polarization in spin-induced improper multiferroics can be generally understood in terms of either the spin current

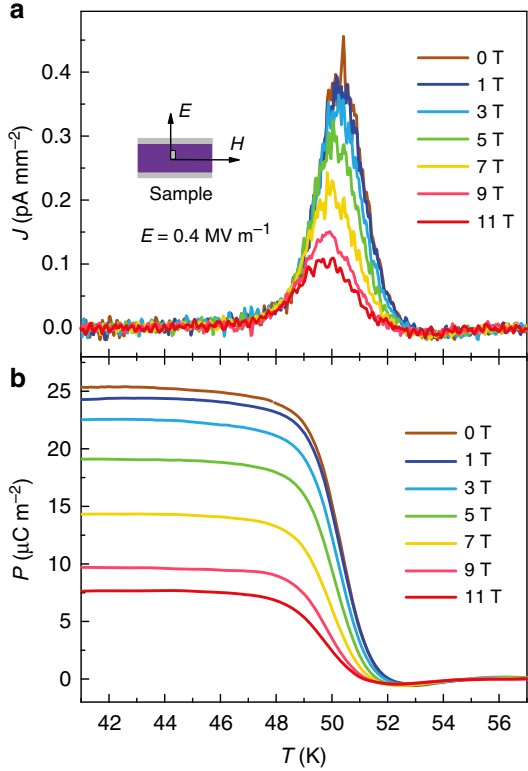

**Fig. 6** Magnetic field control of polarization in the binary perovskite $Mn_2O_3$. **a** The influence of magnetic field on the pyroelectric current. **b** The effect of magnetic field on the electric polarization. The polarization is greatly reduced by applying external magnetic fields

model[45] or inverse Dzyaloshinskii–Moriya (DM) interaction[46]. The extremely complex magnetic structure of the $Mn_2O_3$ perovskite combines both cycloidal and helical components, which generate polarization through different mechanisms. A spin cycloid is a naturally polar object whose polarization is proportional to a double vector product $\mathbf{k} \times (\mathbf{S}_i \times \mathbf{S}_j)$, where the spins $\mathbf{S}_i$ and $\mathbf{S}_j$ rotate, propagating along $\mathbf{k}$[47]. In the case of the $Mn_2O_3$ perovskite, the phase-modulated cycloid combines four distinct sub-components with $\mathbf{k} = \mathbf{k}_0$, $\mathbf{k}_{1+}$, $\mathbf{k}_{1-}$ and $\mathbf{k}_{2+} \equiv \mathbf{k}_{2-}$ generating their own polarizations in the plane perpendicular to the $c$-axis (considering the trigonal approximation for the crystal structure). To the contrary, spin helix is non-polar, unless the crystal structure, on top of which the spin ordering takes place, belongs to a ferroaxial crystal class[27,48]. The trigonal (triclinic) crystal structure of perovskite-type $Mn_2O_3$ indeed satisfies this criterion, implying that the ferroaxial mechanism is relevant as well, and generates polarization along the $c$-axis.

Apparently, the very general character of the magnetic structure is the key ingredient of the great tunability of the electric polarization by applied magnetic field. Significant modifications of the structure, like tilting of the spin plane, do not require symmetry breaking and can gradually occur under external perturbations. This, in turn, changes the relative contributions of the different mechanisms to the polarization, resulting in its strong field dependence.

Thus, the perovskite-type $Mn_2O_3$ represents a unique example of a binary multiferroic oxide that adopts a complex distorted perovskite structure and demonstrates spin-induced ferroelectricity with a strong ME effect. In this material, the $A$-position is fully occupied by the transition-metal cations and, therefore, the role of the $A$-sites is greatly amplified, leading to more fascinating properties that would not be expected in the conventional $ABO_3$

perovskites. The combination of octahedral tilting and deformation, Jahn–Teller distortion, charge/orbital ordering, and multiple magnetic interactions opens up a valuable playground both for designing and engineering of new materials and for investigation of challenging fundamental physical phenomena. Although, bulk samples of the perovskite-type $Mn_2O_3$ were synthesized at HP-HT conditions, the high-pressure perovskite phase can also be stabilized through epitaxial strain in thin films. Fabrication of such binary perovskites at ambient pressure will be an attractive topic in the future.

## Methods

**Sample preparation and characterization.** Fine polycrystalline samples of the perovskite-type $Mn_2O_3$ were synthesized under HP-HT conditions (~20 GPa and 1200–1400 K) using multi-tonne multi-anvil presses at BGI (Universität Bayreuth, Germany)[49]. Details of the synthesis procedures were the same as in previous work[11]. The chemical composition of the samples was examined by scanning electron microscopy (SEM) at a LEO-1530 instrument, microprobe analysis at a JEOL JXA-8200 electron microprobe, and by ICP-MS bulk chemical analysis. The crystal structure of the samples was studied at ambient conditions by means of powder X-ray diffraction (XRD) at high brilliance Rigaku diffractometer. The XRD studies of the sample at different temperatures were carried at a high resolution powder diffraction beamline ID22 (former ID31) at the European Synchrotron Radiation Facility (Grenoble, France) with the wavelength of $\lambda = 0.39993$ Å.

**Magnetic and electric measurements.** DC and AC magnetic susceptibility were measured on a Quantum Design magnetic properties measurement system (MPMS). The temperature and magnetic field dependence of dielectric constant and pyroelectric currents were measured by using a homemade probe on a Cryogen-free Superconducting Magnet System (TeslatronPT, Oxford Instruments). The dielectric constant was measured by using a LCR meter (Aglient 4980A). After poling the sample in an electric field with the cooling process, the pyroelectric current was measured by an electrometer (Keithley 6517B) during the warming process with a constant rate.

**Neutron diffraction measurements.** Two sets of neutron powder diffraction data were collected; one at the ISIS pulsed neutron and muon facility of the Rutherford Appleton Laboratory (UK) on the WISH diffractometer and another at the IBR-2 high-flux pulsed reactor (FNLP Dubna, Russia) on the DN-12 diffractometer. In both cases, samples (~40 mg) were loaded into a cylindrical vanadium cans and measured in the temperature range of 1.5–120 K using an Oxford Instrument Cryostat and 10–200 K using a He-close cycle refrigerator, respectively. Rietveld refinements of the crystal and magnetic structures were performed using the Fullprof program against the data measured in detector banks at average $2\theta$ values of 58°, 90°, 122°, and 154° (WISH) and 45.5° and 90° (DN-12).

**Data availability.** The data that support the findings of this study are available from the corresponding authors upon reasonable request.

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

## Acknowledgements

The work at Institute of Physics was supported by the National Natural Science Foundation of China (Grant No. 11534015 and 51725104), the National Key Research and Development Programm of China (Grant No. 2016YFA0300701), and the Chinese Academy of Sciences (Grants No. XDB07030200) and S.V.O. thanks the Deutsche Forschungsgemeinschaft (DFG, project OV-110/1-3) for the financial support.

## Author contributions

Y.S. and S.V.O. initiated and designed the research. J.C., K.Z., and Y.C. carried out magnetic and electric measurements. D.S. contributed to the analysis of data and discussion. D.D.K., R.D.J., S.E.K., and D.P.K. performed the NPD measurements and analysis, derived magnetic structure models and contributed to manuscript writing. A.M.A. and A.A.T. performed the high-resolution X-ray diffraction studies of the samples at low temperatures, analyzed the XRD data, and contributed to the discussion of the results of the work. X.X. and Z.S. performed the SHG measurements. S.V.O. and L.D. synthesized and characterized the samples and contributed to manuscript writing. J.C. and Y.S. wrote a first draft of the manuscript, and all the co-authors read, revised, and commented on it.

## Additional information

**Competing interests:** The authors declare no competing interests.

