## [Peer Review File · Nature Communications]

Reviewers' comments:

Reviewer #1 (Remarks to the Author):

In this paper, the authors report on the observation of ferroelectricity induced by magnetism in the high-pressure ζ -phase of the binary compound Mn_2O_3 , characterized by a quadruple perovskite $\text{AA}'_3\text{B}_4\text{O}_{12}$ structure. This observation stems from the analysis of data obtained from a wide range of experimental techniques. Specifically, powder x-ray and neutron diffraction and magnetic susceptibility data consistently show a complex spin-density-wave ordering of the Mn ions occupying the A, A' and B sites at $T_1=101$ K, which evolves at $T_2=49$ K. At this temperature, the appearance of a pyroelectric current peak indicates the onset of ferroelectricity, consistent with SHG measurements. In the discussion section, the authors briefly mention the possible mechanisms of ferroelectricity compatible with the observed magnetic structure.

I believe that the experimental study presented in the present paper is of good quality. The observation of magnetic ferroelectricity in a binary transition metal oxide with an unusual perovskite-like structure is interesting, although it should be emphasized that the measured polarizations are small.

For these reasons, the paper may be suitable for publication in Nature Communications. There are however several points in the manuscript that should be improved or clarified.

i) By analyzing the magnetic susceptibility data shown in Fig. 2a, the authors estimate the Weiss temperature to be -373.9 K. The reader should be put in the condition to roughly verify this estimate. To do so, the authors should clearly show the linear extrapolation of the $1/\chi$ curve. In the present plot, the scale of the vertical axis is missing and the origin of the horizontal scale is not clearly indicated as it should. Second, the precision of the above estimate is of the order of a few K (not of 0.1 K), as correctly indicated in Ref. 11, where a quite different value of $-320(5)$ K is given.

ii) The authors should make an effort to make the 3D-visualization of the magnetic structures in Figs. 3b-c easier. In Fig. 3b, the a- and b- axis are not indicated. A projection of the structures on selected crystallographic planes would help. The authors should also discuss the stability of the structures, where both ferro- and antiferro-magnetic interactions are present, e.g. in terms of the Goodenough-Kanamori-Anderson rules.

iii) The analysis and discussion of the pyroelectric current data should be improved.

iii.a. I understand that the broad peak centered around 80 K is related to extrinsic factors, such as charge trapping. To further analyze this important point, the authors should say and show in Fig. S1 how this peak behaves upon reversal of the poling field, as they did for the peak at 49 K in Fig. 4b.

iii.b. How do the authors explain the smaller negative peak centered around 90 K?

iii.c. Because of the broad tail of the pyroelectric current peak at 80 K, it is not obvious to extract the ferroelectricity signal from the background current related to extrinsic effects. Which procedure did the authors adopt to calculate the temperature-dependent polarization shown in Figs. 4,5? In the inset of Fig. 4b, the background current is zero, which is not consistent with the significant background visible in the 40 - 60 K range in Fig. S1.

iii.d. It is customary to perform pyroelectric current measurements at different heating rates in order to confirm the intrinsic origin of the current measured (see, for example, Ref. 37). These

measurements must be carried out in order to strengthen the paper.

iv) As mentioned above, it should be made clear in the text that the observed electric polarizations are small, so a comparison with the ferroelectric properties of related compounds, such as $\text{CaMn}_3\text{Mn}_4\text{O}_{12}$, should be made. How do the authors explain the fact that the polarization found in the present Mn_2O_3 phase is about one order of magnitude smaller than in the above compound? One should compare the values reported on $\text{CaMn}_3\text{Mn}_4\text{O}_{12}$ polycrystals (Ref. 37).

Minor points are as follows:

v) Lines 83-85. It is not clear to me the link between the triclinic symmetry and the fact that the A and B cations would be magnetically and electrically active. This sentence should be rewritten or eliminated.

vi) Lines 213-214. Here, the authors probably mean that "None of the multiferroic binary oxides hitherto studied ... exhibit clear ME effects".

vii) Although several authors call $\text{AA}'_3\text{B}_4\text{O}_{12}$ compounds "A-site ordered perovskites", this name is not appropriate, for a A-site ordering in perovskites does not necessarily lead to the $\text{AA}'_3\text{B}_4\text{O}_{12}$ structure. The name "quadruple perovskites" should be used instead, as it uniquely defines the formula unit and the type of structure as well.

In summary, the present paper may be publishable in Nature Communications if the authors will satisfactorily respond to the above points.

Reviewer #2 (Remarks to the Author):

Cong et al. investigate the multiferroicity of the perovskite manganite Mn_2O_3 , using magnetic susceptibility and electric polarization measurements, and neutron scattering. From these data they revealed the successive magnetic transitions, spiral-like magnetic ground state with ferroelectric polarization. They also observed the field-induced suppression of electric polarization, which correlates with the magnetism.

Though the discovery of rare binary multiferroic will attract attention of experts working on the same field of research, I could not find an aspect which appeals to more general readers. In particular, I got an impression that this multiferroic does not have selling point compared with the others ever discovered such as TbMnO_3 , and binary CuO . In the manuscript, ME functionality in Mn_2O_3 is stressed as a selling point, but I do not think so. Especially, authors mentioned that CuO does not show noticeable ME coupling effects in Line 98. This is not the case. The field-induced polarization change in CuO has been reported by Wang et al. in Nat. Commun. 7, 10295 (2016). Indeed, the strength of the required field is comparable, and the polarization change is larger than that of Mn_2O_3 , and the operation temperature is much higher. I cannot recommend editors accepting this manuscript to Nature Communications.

Reviewer #3 (Remarks to the Author):

In this manuscript, Cong et. al. reported their experimental investigation on binary perovskite manganite Mn_2O_3 as a new binary multiferroic candidate based on multiple measurements. They

suggested that Mn₂O₃ exhibits spin-induced ferroelectricity and magnetoelectric effect at low temperatures. The results can be interesting for the multiferroics community, which will stimulate further studies on related systems. In general, I find this work is scientifically interesting and well done. The experimental results seem to be consistent and the paper was well written. Thus, I think it deserves publication. There are a few small issues needing revisions.

1. I can not find the structural data. The authors should provide them for relevant readers who are interested to do further studies on this material.
2. Fig. 3 (a) should be revised which is obscure, especially for symbols.
3. In the bottom of Fig. 3(b), the down arrows of different sites of Mn²⁺/Mn³⁺/Mn⁴⁺ should be removed (by keeping the color balls only), which may cause misunderstanding.
4. $k \times S_i \times S_j$ should be $k \times (S_i \times S_j)$???

Reply to Reviewers' comments

Reviewer #1 (Remarks to the Author):

In this paper, the authors report on the observation of ferroelectricity induced by magnetism in the high-pressure ζ -phase of the binary compound Mn_2O_3 , characterized by a quadruple perovskite $AA_3B_4O_{12}$ structure. This observation stems from the analysis of data obtained from a wide range of experimental techniques. Specifically, powder x-ray and neutron diffraction and magnetic susceptibility data consistently show a complex spin-density-wave ordering of the Mn ions occupying the A, A' and B sites at $T_1=101$ K, which evolves at $T_2=49$ K. At this temperature, the appearance of a pyroelectric current peak indicates the onset of ferroelectricity, consistent with SHG measurements. In the discussion section, the authors briefly mention the possible mechanisms of ferroelectricity compatible with the observed magnetic structure.

I believe that the experimental study presented in the present paper is of good quality. The observation of magnetic ferroelectricity in a binary transition metal oxide with an unusual perovskite-like structure is interesting, although it should be emphasized that the measured polarizations are small.

For these reasons, the paper may be suitable for publication in Nature Communications. There are however several points in the manuscript that should be improved or clarified.

Reply:

We thank the reviewer for his/her positive assessment on our work: "The observation of magnetic ferroelectricity in a binary transition metal oxide with an unusual perovskite-like structure is interesting". Following the reviewer's comments/suggestions, we have accordingly revised the manuscript.

i) By analyzing the magnetic susceptibility data shown in Fig. 2a, the authors estimate the Weiss temperature to be -373.9 K. The reader should be put in the condition to roughly verify this estimate. To do so, the authors should clearly show the linear extrapolation of the $1/\chi$ curve. In the present plot, the scale of the vertical axis is missing and the origin of the horizontal scale is not clearly indicated as it should. Second, the precision of the above estimate is of the order of a few K (not of 0.1 K), as correctly indicated in Ref. 11, where a quite different value of -320(5) K is given.

Reply:

We thank the reviewer for pointing out this problem. We have re-plotted Fig. 2a and re-estimated the Weiss temperature. Now the value is close to that estimated in Ref. 11 in which the magnetization was measured up to 380 K in 5 T, and so, the Weiss temperature determination seems to be more accurate.

ii) *The authors should make an effort to make the 3D-visualization of the magnetic structures in Figs. 3b-c easier. In Fig. 3b, the a- and b- axis are not indicated. A projection of the structures on selected crystallographic planes would help. The authors should also discuss the stability of the structures, where both ferro- and antiferro-magnetic interactions are present, e.g. in terms of the Goodenough-Kanamori-Anderson rules.*

Reply:

Following the reviewer's comment, we have added two new Figures displaying the details of the magnetic structures we determined, in Supplementary Information (Figures S1 and S2). The isotropic Heisenberg exchange interactions stabilizing complex magnetic structures in $A^{2+}Mn_7O_{12}$ manganites were discussed in several previous publications. This approach is relevant for the case of the perovskite Mn_2O_3 too. We have added an appropriate discussion with references and explanation of the main distinct features of the perovskite Mn_2O_3 in the revised manuscript, as follows:

“The factors stabilizing the complex magnetic structures in the binary perovskite Mn_2O_3 are probably common for the whole series of $A^{2+}Mn_7O_{12}$ manganites [28,29,32]. The central role is played by competing exchange interactions of Mn located in both the A-site and B-site perovskite positions, modulated by orbital density wave. The main distinctive feature of Mn_2O_3 is the moments direction in the high-temperature spin density wave and the cycloidal component in the ground state, both implying different type of magnetic anisotropy, whose origin requires further experimental and theoretical studies.”

Note, that consideration of the exchange interactions on a basis of the Goodenough-Konamori rules would require a detailed information about the orbital density wave. This information could be obtained only in additional extensive studies which are beyond the scope of the present work.

iii) *The analysis and discussion of the pyroelectric current data should be improved.*

iii.a. I understand that the broad peak centered around 80 K is related to extrinsic factors, such as charge trapping. To further analyze this important point, the authors should say and show in Fig. S1 how this peak behaves upon reversal of the poling field, as they did for the peak at 49 K in Fig. 4b.

Reply:

Following this comment, we have included Figure. S3 displaying the pyroelectric current data with negative poling field. A supporting discussion has been added.

iii.b. *How do the authors explain the smaller negative peak centered around 90 K?*

Reply:

It is not clear what causes this small negative peak in the pyroelectric current. This feature was also reported in the pyroelectric current of $CaMn_7O_{12}$ (ref. 35). As the

high-temperature pyroelectric peak is ascribed to extrinsic factors, it does not affect the discussion and conclusion of our paper.

iii.c. Because of the broad tail of the pyroelectric current peak at 80 K, it is not obvious to extract the ferroelectricity signal from the background current related to extrinsic effects. Which procedure did the authors adopt to calculate the temperature-dependent polarization shown in Figs. 4, 5? In the inset of Fig. 4b, the background current is zero, which is not consistent with the significant background visible in the 40-60 K range in Fig. S1.

Reply:

In order to obtain the intrinsic polarization value from the pyroelectric current, we have to subtract the background current. As the reviewer commented, this information was not clearly presented. In the revised manuscript, we have described in more detail the procedure of the background current subtraction in Fig. S6 in Supplementary Information.

iii.d. It is customary to perform pyroelectric current measurements at different heating rates in order to confirm the intrinsic origin of the current measured (see, for example, Ref. 37). These measurements must be carried out in order to strengthen the paper.

Reply:

Following the reviewer's comment, we have included the pyroelectric data measured with different heating rates in Fig. S4 in Supplementary Information. The pyroelectric peak does not shift with the heating rate, thereby further confirming the intrinsic spin-induced ferroelectricity.

iv) As mentioned above, it should be made clear in the text that the observed electric polarizations are small, so a comparison with the ferroelectric properties of related compounds, such as $\text{CaMn}_3\text{Mn}_4\text{O}_{12}$, should be made. How do the authors explain the fact that the polarization found in the present Mn_2O_3 phase is about one order of magnitude smaller than in the above compound? One should compare the values reported on $\text{CaMn}_3\text{Mn}_4\text{O}_{12}$ polycrystals (Ref. 37).

Reply:

The large value of polarization reported in $\text{CaMn}_7\text{O}_{12}$ (Ref. 27 and 37) was due to the broad and high pyroelectric current peak around 70-90 K, similar to that observed in the perovskite Mn_2O_3 . If we integrate this high-temperature peak, we will also get a large value of polarization for the perovskite Mn_2O_3 . However, Terada et al. performed a systematic investigation and concluded that those reported large polarization values in $\text{AMn}_7\text{O}_{12}$ are not associated with intrinsic ferroelectricity (Ref. 35). In the present study, we additionally performed the SHG measurements. The SHG data further clarify that the high-temperature phase above ~50 K is not ferroelectric. Therefore, the reported large polarization value for $\text{CaMn}_7\text{O}_{12}$ is not intrinsic.

Minor points are as follows:

v) Lines 83-85. *It is not clear to me the link between the triclinic symmetry and the fact that the A and B cations would be magnetically and electrically active. This sentence should be rewritten or eliminated.*

Reply:

We have removed the phrase “by virtue of the triclinic symmetry”.

vi) Lines 213-214. *Here, the authors probably mean that “None of the multiferroic binary oxides hitherto studied ... exhibit clear ME effects”.*

Reply:

We have removed this statement.

vii) *Although several authors call AA’3B4O12 compounds “A-site ordered perovskites”, this name is not appropriate, for a A-site ordering in perovskites does not necessarily lead to the AA’3B4O12 structure. The name “quadruple perovskites” should be used instead, as it uniquely defines the formula unit and the type of structure as well.*

Reply:

Following the reviewer’s comment, we have replaced “A-site ordered perovskites” by “quadruple perovskites” in the revision.

In summary, the present paper may be publishable in Nature Communications if the authors will satisfactorily respond to the above points.

Reviewer #2 (Remarks to the Author):

Cong et al. investigate the multiferroicity of the perovskite manganite Mn2O3, using magnetic susceptibility and electric polarization measurements, and neutron scattering. From these data they revealed the successive magnetic transitions, spiral-like magnetic ground state with ferroelectric polarization. They also observed the field-induced suppression of electric polarization, which correlates with the magnetism.

Though the discovery of rare binary multiferroic will attract attention of experts working on the same field of research, I could not find an aspect which appeals to more general readers. In particular, I got an impression that this multiferroic does not have selling point compared with the others ever discovered such as TbMnO3, and binary CuO. In the manuscript, ME functionality in Mn2O3 is stressed as a selling point, but I do not think so. Especially, authors mentioned that CuO does not show noticeable ME coupling effects in Line 98. This is not the case. The field-induced polarization change in CuO has been

reported by Wang *et al.* in *Nat. Commun.* 7, 10295 (2016). Indeed, the strength of the required field is comparable, and the polarization change is larger than that of Mn_2O_3 , and the operation temperature is much higher. I cannot recommend editors accepting this manuscript to *Nature Communications*.

Reply:

We thank the reviewer for reminding us the recent progress in the binary multiferroic CuO under high magnetic fields. We have cited this paper and revised the context in the discussion.

Compared to CuO, the perovskite-type Mn_2O_3 represents a unique example of binary oxide adopting the perovskite structure. As pointed out by reviewer #1, the observation of spin-induced ferroelectricity in a binary transition metal oxide with an unusual perovskite-like structure is interesting. Our paper is the first one which is reporting the multiferroicity in a binary perovskite. Furthermore, using a variety of experimental techniques, we clarify the important controversy concerning the giant ferroelectricity, which is often reported for quadruple perovskites of this family, like $CaMn_7O_{12}$. Our results for the perovskite Mn_2O_3 , which can be considered as a parent compound for this family, demonstrate the absence of ferroelectricity between T1 and T2 transition points for this perovskite family. Thus, the previously reported giant ferroelectricity is delusive, and only the moderate ferroelectricity below T1 is intrinsic. This fact settles a long debate and constitutes an important result of general interest for researchers dealing with both perovskite materials and ferroelectric properties.

Reviewer #3 (Remarks to the Author):

In this manuscript, Cong et. al. reported their experimental investigation on binary perovskite manganite Mn_2O_3 as a new binary multiferroic candidate based on multiple measurements. They suggested that Mn_2O_3 exhibits spin-induced ferroelectricity and magnetoelectric effect at low temperatures. The results can be interesting for the multiferroics community, which will stimulate further studies on related systems. In general, I find this work is scientifically interesting and well done. The experimental results seem to be consistent and the paper was well written. Thus, I think it deserves publication. There are a few small issues needing revisions.

Reply:

We thank the reviewer for his/her positive assessment of our work and recommendation for publication.

1. I can not find the structural data. The authors should provide them for relevant readers

who are interested to do further studies on this material.

Reply:

The structural information of the Mn_2O_3 perovskite has been described on page 4 and 6 in the manuscript. Detailed structural information for the perovskite Mn_2O_3 was presented in previous publication [S. V. Ovsyannikov, *et al. Angew. Chem. Int. Edit.* **52**, 1494-1498 (2013), in Supplementary Information file]. In the present work we referred to this publication.

2. *Fig. 3 (a) should be revised which is obscure, especially for symbols.*

Reply:

Following the reviewer's comment, we have divided Fig. 3a-c into two figures (new Figs. 3 and 4), and now the details of each panel are more clearly seen.

3. *In the bottom of Fig. 3(b), the down arrows of different sites of $\text{Mn}^{2+}/\text{Mn}^{3+}/\text{Mn}^{4+}$ should be removed (by keeping the color balls only), which may cause misunderstanding.*

Reply:

Following the reviewer's comment, we have modified the down arrow labels in Fig. 3b (new Fig. 4).

4. *$k \times S_i \times S_j$ should be $kx(S_i S_j)$???*

Reply:

This is a double vector product based on the inverse Dzyaloshinskii-Moriya (DM) interaction. Both versions are correct. We can use the brackets as the reviewer suggested.

REVIEWERS' COMMENTS:

Reviewer #1 (Remarks to the Author):

In the revised version, the authors have suitably responded to all points raised in my first report. I therefore believe that the paper is now suitable for publication in Nature Communications.

The authors should correct the following minor points:

1. In the figure caption of Fig. 4, the vector P must be defined.
2. In the paragraph starting at line #223, it seems to me that the last word 'is' should be removed.

Reviewer #3 (Remarks to the Author):

The authors have addressed all my comments. I can recommend the acceptance now.

Responses to reviewers' comments

Reviewer #1 (Remarks to the Author):

In the revised version, the authors have suitably responded to all points raised in my first report. I therefore believe that the paper is now suitable for publication in Nature Communications.

The authors should correct the following minor points:

1. In the figure caption of Fig. 4, the vector P must be defined.

Response:

We have defined the vector P in the figure caption of Fig. 4.

2. In the paragraph starting at line #223, it seems to me that the last word 'is' should be removed.

Response:

We have deleted the word "is".